# Characterization of PPS Piston and Packing Ring Materials for High-Pressure Hydrogen Applications

**DOI:** 10.3390/polym16030412

**Published:** 2024-02-01

**Authors:** Alexander Pöllinger, Julia Maurer, Thomas Koch, Stefan Krenn, Bernhard Plank, Sabine Schwarz, Michael Stöger-Pollach, Eleni Siakkou, Karolina Smrczkova, Michael Schöbel

**Affiliations:** 1Leobersdorfer Maschinenfabrik GmbH, 2544 Leobersdorf, Austria; 2Research Group Computer Tomography, University of Applied Sciences Upper Austria, 4600 Wels, Austria; 3Institute of Materials Science and Technology, Technische Universität Wien, 1040 Vienna, Austria; 4AC2T Research GmbH, 2700 Wiener Neustadt, Austria; 5Service Center for Electron Microscopy (USTEM), Technische Universität Wien, 1040 Vienna, Austria; 6MOCOM Compounds GmbH & Co. KG, 20539 Hamburg, Germany; 7X-ray Center, Technische Universität Wien, 1040 Vienna, Austria

**Keywords:** fiber-reinforced polymers, hydrogen technology, thermo-mechanical properties, friction and wear, visco-elastic deformation, X-ray imaging, transmission electron microscopy

## Abstract

The widespread adoption of renewable energy hinges on the efficient transportation of hydrogen. Reciprocating piston compressor technology in non-lubricated operation will play a key role, ensuring high flow rates and compression ratios. These systems rely on advanced high-strength sealing solutions for piston and rod packing rings utilizing advanced fiber-reinforced polymers. Polyphenylene sulfide (PPS) polymer matrix composites have seen use in tribological applications and promise high mechanical strength and wear resistance. The presented work describes carbon and glass fiber-reinforced PPS matrix polymers in comparison, which are characterized by complementary methods to investigate their properties and potential for application in reciprocating compressor under non-lubricated operation. Thermo-mechanical and tribological testing was supported by microstructure analysis utilizing advanced X-ray and electron imaging techniques. New insights in micromechanical deformation behavior in regard to fiber materials, interface strength and orientation in fiber-reinforced polymers are given. Conclusions on the suitability of different PPS matrix composites for high-pressure hydrogen compression applications were obtained.

## 1. Introduction

The increasing utilization of renewable power generation and associated production volatility will necessitate the application of scalable energy storage solutions. Gaseous hydrogen in power-to-X applications will serve as a key technology to alleviate these challenges, as existing infrastructure can be re-used and energy can be stored over long periods of time [1]. The low volumetric energy density of hydrogen demands pressures exceeding 350 bar for applications such as the mobility sector where compact storage solutions are necessary [2]. This poses engineering challenges to reciprocating compressors, as high gas quality requirements demand non-lubricated operation. Piston and packing rings are tribological systems under high pressure–velocity loading and demand high-performance fiber-reinforced polymer materials to achieve adequate sealing ring life.

Today’s application of hydrogen-reciprocating compressors can be found at discharge pressures of up to 250 bar utilizing oil-lubricated cylinders and multiple compression stage setups to achieve the pressure targets [3]. These systems include piston and packing ring solutions consisting of polytetrafluoroethylene (PTFE) matrix composites with excellent tribological properties combined with carbon fiber (CF) reinforcements to increase structural stiffness and improve long-term stability [4,5]. Discharge pressures of 500 bar cannot be achieved with sufficient ring life, as failure modes such as ring extrusion or excessive wear are observed [6].

Non-lubricated cylinders are significantly more challenging to design, as the requirements on packing and piston ring material, in regard to thermal stability, abrasion behavior and mechanical strength, set high demands on the ring materials [7]. High yield strengths, low wear rates and stable friction coefficients are necessary with respect to the intended operating conditions.

Consequentially, high-performance composite materials with polymer matrices such as PEEK are seeing increasing usage as seen in recent applications [8,9,10].

PPS polymers and composites have good thermal stability, chemical resistance and exhibit reliable performance in hostile environments including hydrogen atmospheres [11,12,13,14,15]. Therefore, PPS has seen a wide range of application including electronics, electrical devices, vehicle parts and aerospace components [16,17,18].

Hence, PPS is a promising candidate material for piston and packing ring solutions in high pressure although pure PPS suffers from high friction and wear [19,20]. The addition of solid lubricants can significantly improve tribological performance and promises potential for high-pressure hydrogen reciprocating compressors [21,22]. Previous work has shown that the performance of a polymer–metal tribo-pair is highly sensitive to the growth of a self-lubricating film on the surface of the metal counterpart. This effect can reduce friction and improve wear resistance by preventing direct contact of the sliding surfaces and is one of the main mechanisms defining the excellent properties of PTFE matrix composites [23]. Hence, the experiments discussed in this work are conducted on PPS matrix composites with glass fiber and carbon fiber reinforcements with and without PTFE as a solid lubricant additive.

The design of this study aims to address the existing research gap by investigating the thermo-mechanical properties and microstructural effects of four experimental PPS matrix composites on wear resistance. Specifically, we explore the potential of incorporating solid lubricants and additives like PTFE and different reinforcement fiber types to enhance the composite materials’ performance. The goal of this work is to determine the thermo-mechanical properties and effects of the microstructure on the wear resistance of the experimental PPS matrix composites in comparison, and the investigations strive to present a comparative study to determine the influence of the above-presented additive phases on the composites’ properties.

Finally, the investigations aim to present a comparative study to determine the composites’ suitability as piston ring and piston rod packing material under conditions for high-pressure hydrogen compressors and identify potential candidate materials for real-world applications.

## 2. Materials and Methods

### 2.1. Test Materials

Injection-molded sample plaques of four commercially available PPS compounds with 30 wt.% glass fiber (GF) reinforcement and 30 wt.% carbon fiber (CF) reinforcement, respectively, were supplied by MOCOM Compounds GmbH & Co. KG (Hamburg, Germany). The composition of the PPS compounds supplied is summarized in Table 1.

The neat PPS polymer had an average Young´s modulus of 4 GPa, a tensile stress at break range of 30 MPa–90 MPa, a coefficient of linear thermal expansion (CTE) average of 50 ppm/K and a density of 1.35 g/${\mathrm{cm}}^{3}$. The GF reinforcement consisted of chopped E-glass fiber with a nominal diameter of 10 μm, a nominal chop length of 4.5 mm, an ultimate tensile strength of 3450 to 3790 MPa, a Young´s modulus of 72.4 GPa and a density of 2.6 g/${\mathrm{cm}}^{3}$. The CF blend contained chopped carbon fiber with a filament diameter of 7 μm, a nominal fiber length of 6 mm, a single filament strength of 4.0 GPa, a single filament modulus of 240 GPa and a density of 1.8 g/${\mathrm{cm}}^{3}$. One sample of each fiber type was additionally modified with 15 wt.% of a PTFE powder lubricant additive that had a particle size mean value of 12 μm and a density of 2.15 g/${\mathrm{cm}}^{3}$. The wt.% contents of CF and GF were kept the same for the lubricated materials, only reducing the overall PPS matrix mass fraction in the mixture to accommodate for the PTFE.

The blends were prepared at MOCOM using a co-rotating twin screw extruder with a strand pelletizing system in order to obtain granules which could then be injection molded into rectangular sample plaques with the dimensions of 80 × 80 × 3 mm. Prior to injection molding, the material was pre-dried in a dry-air drier at 130 to 140 °C for 2 to 4 h. Mass temperature during injection molding was 320 to 340 °C, and the tool temperature was kept at > 140 °C.

### 2.2. X-ray Computed Tomography

X-ray Computed Tomography (CT) scans were performed for the evaluation of the internal microstructure and the fracture behavior. The CT scans were performed at the University of Applied Sciences Upper Austria (Wels, Austria) using two different laboratory CT devices. Glass fiber-reinforced PPS samples (with and without PTFE particles) were milled out from a 3 mm thick plate, resulting in a cross-section of 3 × 3 $mm^{2}$ and a length of 20 mm. The scans were conducted with a Nanotom 180 NF device (phoenix|X-ray, GE Sensing & Inspection Technologies GmbH, Wunstorf, Germany) with a voxel edge length of 2 μm. The CT device was equipped with a tungsten on diamond target, and the sample was scanned at a voltage of 80 kV. A total of 1800 projections were recorded with an integration time of 600 ms and average of 3. For the evaluation of the fiber distribution, the in-house developed fiber characterization pipeline was used [24,25]. Scans with higher resolution were performed on the fracture surface of the miniaturized test specimens (region of interest shown in Figure 1a) of the carbon fiber-reinforced PPS (with and without PTFE particles). The overview scan of the fracture surface of these CFRPs (cross-section approximately 1.6 × 1.6 $mm^{2}$) was also performed on the Nanotom device at a voxel edge length of 1 μm. The device was operated with the molybdenum target on a small focal spot (Mode 2) and a voltage of 55 kV. Again, 1800 projections were gained, with an integration time of 1200 ms and average of 9. Afterwards, the sample was sanded to a smaller cross-section using wetted SiC sanding paper, enabling a higher resolution scan of a selected fracture zone on the Easytom CT device (RX solutions, Chavanod, France) equipped with an lanthanhexaborid (LaB_6_) target. This scan was performed at a voxel edge length of 0.5 μm and with 60 kV. A total of 4000 projections with an integration time of 1000 ms and average of 3 were acquired. The CT measurement setups are summarized in Table 2.

### 2.3. Tribometry

The fiber-reinforced PPS polymers were tribologically tested to compare the friction and wear behavior and the influence of glass and carbon fiber content with and without PTFE addition. Testing was performed at AC2T research GmbH (Wiener Neustadt, Austria) using the oscillating tribometer 1 [27] with a pin on plate configuration as schematically illustrated in Figure 2a and a contact pressure 24.8 MPa at a pin velocity $v_{p}$ of 0.1 m/s. The PPS pins had a cylindrical shape with a diameter of 4 mm and were loaded with a normal force of 312 N to achieve the required constant contact pressure normal to the counter surface and moving direction of the simulated tribo-system. The ground plate material was made of Boehler W720 steel in solution heat-treated condition [28] and sanded to an initial surface roughness of Ra=0.16 μm, which was the same surface topology as used in compressor cylinder liners. The chemical composition of the steel counter surface is summarized in Table 3.

The stroke of the oscillating tribometer was set to 50 mm operated at 1 Hz, resulting in the required average pin velocity. Operation under bone-dry atmosphere was ensured by N_2_ gas of purity grade N5.0 flooding of the sample environment during testing. The measurements were carried out during 3000 cycles to obtain stable and representative results for operation. Analysis of the wear track and pin using a 3D microscope (Alicona InfiniteFocus) before and after testing was followed by a differential topography analysis as well as gravimetrically measurement, which delivered reliable data for wear as a function of time.

### 2.4. Transmission Electron Microscopy

Transmission electron microscopy (TEM) supported by focused ion beam (FIB) milling was performed to analyze the material depositions and wear mechanisms on the counter surfaces as a function of depth. FIB target preparation was conducted on Boehler W720 solution heat treated counter surfaces in the wear track after tribological testing with PPS polymers under comparable conditions. From the center of the 4 mm wide wear tracks, TEM sample lamellae of a dimension of 10 μm in depth, 20 μm in width and an initial thickness of 3 μm were cut perpendicular to the oscillation direction using a Thermofisher Scios 2 Dual Beam FIB operated initially at 30 kV and 30 nA. Prior to cutting, a protective platinum Pt layer was deposited on top of the wear track to avoid damage and loss of the sensitive polymer layer by sample manipulation and beam damage. After extraction, the sample was fixed to a manipulator tool and subsequently thinned from 3 μm to 80 nm final thickness by additional ion milling. A finalizing gentle milling preparation step was performed in the FIB with low ion beam energies and currents ( 3 kV, 16 pA) to remove surface contamination and the remaining beam damage induced by initial cutting. A field emission transmission electron microscope FEI Tecnai F20 was used for analysis. The TEM was operated at 200 kV for imaging of the FIB lamellae in the direction of the wear track as a function of depth. The electron transparent lamellae were fixed on copper rings and positioned with a double-tilt sample holder within the center of the electron beam of the microscope. A Gatan Rio16 CCD camera was used for bright field (BF) and dark field (DF) imaging below the sample in the column of the microscope. Electron energy loss spectroscopy (EELS) was used for highly sensitive element analysis, which was acquired by a Gatan GIF Tridiem detector unit positioned below the conventional detector benches.

### 2.5. Dynamic Mechanical Analysis

The storage modulus ($E^{\prime }$) of the PPS polymers was measured using dynamic mechanical analysis (DMA) according to ISO 6721-5 utilizing a TA Instruments DMA Q800 in 3-point bending mode with a support span of 50 mm. The specimens were prepared by cutting from the center region of the injection-molded plates with the main axis oriented in the injection direction to a length of 60 mm and a width of 8 mm. The storage modulus was measured in a temperature range of −120 to 290 °C and a heating rate of 3 K/min. A dynamic amplitude of 0.05% and a frequency of 1 Hz at a ratio of static to dynamic load of 1.25 was applied during measurement.

### 2.6. Shear Testing

To determine the resistance of the candidate materials’ capabilities to withstand ring extrusion in high-pressure applications, the shear strength was determined using a punch tool according to ASTM D 732 [30] For that, the 10.1 mm center hole was carefully drilled into the injection-molded plates with a lathe. A Zwick Roell Z050 testing machine was used at 1.25 mm/min crosshead speed, and 2 tests on specimen taken from different sample plaques were performed that were representative of each material type presented in this work. Dividing the maximum force $F_{max}$ by the area of the sheared edge gives a shear strength value according to the equation
(1)$$\tau_{max}=\frac{F_{max}}{d\pi t},$$
with the diameter of the punch d=25.4 mm and the thickness of the plates t=3 mm.

### 2.7. Thermal Expansion

Dilatometry tests were conducted on PPS polymer materials to measure the length change as a function of time and calculate the linear coefficient of thermal expansion in the operation temperature range of a compressor. The specimens were cut with a Struers Accutom 100 to a size of 3 × 3 × 15 $mm^{3}$ and water cooled to avoid overheating. A TA Instruments TMA Q400 was used for thermal expansion measurements in a temperature range of −50 to 300 °C and a heating rate of 2 K/min. On the large sample dimension of 15 mm, the length was recorded as a function of time.

### 2.8. Standard Tests

Light optical micrography (LOM) was performed using a Zeiss Axio Imager M2m light optical microscope on cut and subsequently polished sample cross-sections. A Zwick Roell Z 050 test machine equipped with a 2.5 kN load sensor was used for tensile testing on type 5A specimens according to ISO 527-2 [31], with a crosshead speed of 10 mm/s at room temperature (23 °C). Test conditions at ambient temperature were chosen to investigate the relative changes in material properties due to the differences in composition. Each of the 2 specimens per material tested, taken from different individual sample plates but in the same orientation/position, were water cut from the center region of the injection-molded plates with the specimen’s main axis oriented in the injection direction, as shown in Figure 1b. A EMCO M1C 010 test setup was used for Vickers hardness testing with 30 s holding time.

## 3. Results

In this section, the characteristics of the samples (Table 1) are described by each test method, respectively.

### 3.1. Microstructure

The orientation and distribution of the composites’ constituents after injection molding are significant to the performance parameters of the polymer in piston and packing ring application. Hence, an assessment of the microstructure was conducted by analysis of the LOM images depicted in Figure 3.

All four images were taken from the center of the specimen plates normal to the direction of tensile testing, as shown in Figure 1b. Each material shows indications of normal to cross-section-oriented heterogeneous fiber distribution. As the microsection indicates an alignment of fiber orientation, which is expected for the injection-molding part, the importance of assessing the manufacturing process is demonstrated to tailor the mechanical properties for specific engineering applications. The slightly greater diameter of the glass fibers within PPS/GF30f and PPSPTFE/GF30f is visually apparent when compared to the carbon fibers found in PPS/CF30f and PPSPTFE/CF30f. Furthermore, the observed difference in the fiber volume fraction between fiber types originates from the method of compounding, based on mass fraction, and can be explained by the density differences between GF and CF.

The fiber–matrix interface plays a pivotal role in defining the properties of heterogeneous systems, such as fiber-reinforced polymer composite materials [26,32]. PTFE is visible as darker spots in both PPSPTFE/GF30f (Figure 3c) and PPSPTFE/CF30f (Figure 3d).

The heterogeneous fiber distribution seen in the micrographs is confirmed by analysis of tomography data. Figure 4 shows the main fiber orientation tensor axis gathered for PPS/GF30f and PPSPTFE/GF30f along the thickness of the injection-molded sample plates. The orientation of the coordinate system is chosen according to Figure 1b. Both materials exhibit three distinct zones with an overall plane fiber distribution. In the skin layers at the top and bottom of the work piece, fibers predominantly oriented in the injection-molding direction are observed at the top and bottom surfaces of the work piece. This phenomenon arises from the elongation forces generated during fountain flow at the melt front as well as the resulting shear flow occurring in the cavity [33]. In contrast, glass fibers in the center of the plate are shown to be predominantly normal to the inflow direction.

The comparison of micrograph images between PPS/GF30f and PPS/CF30f, presented in Figure 5, indicates similar fiber orientations for both glass and carbon fiber materials. Due to there being equal fractions of the parallel and perpendicular-oriented fibers with respect to the injection-molding direction, tensile and shear test specimens were cut into each molding direction but are representative of both fiber orientations.

### 3.2. Mechanical Testing

Tensile, shear and hardness testing was conducted to analyze the changes of mechanical properties caused by variation of fiber type and the addition PTFE to the PPS matrix with the results displayed in Figure 6a,b. The samples were cut in-plane and in full thickness from the injection-molded plates. The tensile tests show that the glass fiber-reinforced materials exhibit higher fracture elongation than the CF-containing counterparts. The addition of PTFE to the GF-reinforced material PPSPTFE/GF30f unintuitively does not have a significant influence on the stiffness behavior in tensile testing, and an increase in ultimate tensile strength from 113.2 to 123.5 MPa as well as improved fracture elongation can be observed.

PPS/CF30f shows the highest ultimate tensile strength of 153.3 MPa and a low fracture elongation of 1%. The incorporation of PTFE has a significant adverse effect on tensile strength reaching only 112.3 MPa for PPSPTFE/CF30f, albeit at a 30% higher maximum strain. The reduction of material strength has been observed in previous work and is attributed to PTFE situated at the fiber–matrix interface leading to a weakening of the composite interface strength [26].

Contrary to the tensile tests, the shear tests show no significant difference between the fiber types. Both PPS/GF30f and PPS/CF30f achieve shear strengths of approximately 85 MPa. The addition of PTFE has an adverse effect on the ultimate shear strength. The results of the shear test show a distinctive drop in shear strength with PTFE for both fiber types. PPSPTFE/GF30f achieved 65.7 MPa, while PPSPTFE/CF30f showed a slightly higher shear strength of 67 MPa. A summary of the mechanical tests is presented in Table 4.

The hardness behavior, summarized in Table 5, shows 33 HV reached by PPS/CF30f followed by the glass fiber-reinforced PPS/GF30f at 29 HV. In comparison, reduced hardness with a PTFE content of 27 HV can be observed in PPSPTFE/CF30f, as well as PPSPTFE/GF30f reaching only 25 HV; however, this is consistent with the trend of CF-reinforced PPS exhibiting higher hardness than specimens with GF.

### 3.3. Thermal Properties

The results of DMA testing are illustrated in Figure 7b showing temperature-dependent storage moduli ($E^{\prime }$) for all four polymers in comparison. The PTFE-containing materials (PPSPTFE/GF30f and PPSPTFE/CF30f) show a step in the temperature region from 15 to 35 °C, which is related to the crystalline phase transition of PTFE. At 19 °C, the order changes from a well-ordered triclinic phase II to a partially ordered hexagonal phase IV. Above 30 °C, it changes to phase I, which is a pseudohexagonal structure with dynamic conformational disorder [34].

All four specimens show a nearly linear decline in storage modulus in the reciprocating compressor operating temperature of 35 to 90 °C with similar rates, but the comparison of the different reinforcement types clearly indicated an overall higher storage modulus for the CF fiber materials (PPS/CF30f, PPSPTFE/CF30f) compared to GF-reinforced PPS (PPS/GF30f, PPSPTFE/GF30f).

PPS/CF30f has the highest storage modulus at room temperature of 22.9 GPa followed by PPSPTFE/CF30f with 18.6 GPa, which equals a reduction of 19%. Surprisingly, the addition of PTFE does not show such a strong effect on the glass fiber-reinforced PPS, with PPS/GF30f reaching 10.8 GPa and PPSPTFE/GF30f at 10.3 GPa, respectively. Over 100 °C, a steep decline in storage modulus is measured for all four PPS composite specimens.

During dilatometry testing, the dimension change was measured continuously for each PPS composite in a temperature range of −50 to 250 °C, as shown in Figure 7b. Contrary to the DMA testing, in the compressor operating range of 35 to 90 °C, fiber type does not have a strong affect on thermal expansion. Both PPS/GF30f and PPS/CF30f exhibit a similar CTE of approximately 11 ppm/K. The introduction of 15 wt.% PTFE has a significant effect on the thermal expansion properties of the composite. Throughout the whole temperature range of the test, both PPSPTFE/GF30f and PPSPTFE/CF30f show linear thermal expansion at an increased rate compared to the non-lubricated materials of 21 ppm/K and 23 ppm/K, respectively. As an important design parameter for reciprocating compressor piston and packing rings, Table 6 gives an overview of the calculated CTE in the compressor operating range of 35 to 90 °C.

Contrary to the CTE results, the mechanical properties show the comparison between GF and CF adheres to the rule of mixture for the tensile and DMA tests. Calculated fiber volume fractions for the composites and the lower Young’s modulus of GF can predict the unfavorable thermo-mechanical properties of PPS/GF30f versus PPS/CF30f.

### 3.4. Fracture Analysis

Characterization of failure mechanisms and micromechanical deformation on the PPS composite materials was conducted by analysis of tomographic imaging of the fracture surfaces (Figure 8 and Figure 9) produced during tensile testing with the objective of gaining better insight into the reinforcing/weakening mechanisms of GF/CF fibers and PTFE lubricant.

The tomography of PPS/GF30f indicates fiber pullout and matrix cracking as the main failure mechanisms. Crack propagation cuts through the PPS phase from fiber to fiber without any crack branching in the matrix material (Figure 8a). A similar conclusion can be drawn for PPSPTFE/GF30f, albeit signs of crack branching can be observed. The tensile test specimen PPS/CF30f and PPSPTFE/CF30f fracture surfaces analyzed by tomography are shown in Figure 9. Analogous to PPS/GF30f, PPS/CF30f shows matrix cracking between the fibers and fiber pullout. The addition of PTFE promotes a different crack propagation which can be observed in the ROI of Figure 9b, as presented in Figure 9c. PTFE particles distributed in the matrix lead to deflection, branching of the cracks, and interface delamination at the fibers, as seen in previous work [26].

Micrographs of the shear tested samples, presented in Figure 10, give insight into the fracture behavior of the PPS materials under shear loading. Both PPS/GF30f and PPS/CF30f show the typical structures of a brittle fracture. In contrast, the PTFE added materials (PPSPTFE/GF30f and PPSPTFE/CF30f) exhibit a higher degree of plastic deformation, which is indicated by the visible displacement of the fibers in the direction of the load.

### 3.5. Tribological Experiments

The assessment of the tribological properties of the PPS composite specimen characterized in this study is essential to develop suitable compressor ring materials, as excessive material abrasion is a main cause for premature failure. The results of the tests conducted in this study are summarized in Figure 11, showing coefficient of friction (CoF) and in situ wear measurements during tribometer test runs for each material. Additionaly to each test result indicated by a colored line, averages and standard deviations for CoF and wear are displayed in black and lighter colored surfaces, respectively.

The PPS composites without PTFE (PPS/GF30f, PPS/CF30f) show excessive wear with a short run-in phase and a high subsequent wear rate. Additionally, PPS/GF30f exhibits a sharp increase in wear rate, which is attributed to the overheating of the PPS pin material after approximately 1100 cycles, although this is not exceeding the wear of PPS/CF30f. The latter also has a degressive wear rate at the end of the test run, indicating an improvement in tribological performance, although this is still exceeding the wear rates of all other materials tested in this study. High coefficients of friction can also be observed for the non-lubricated materials, where the increase in wear for PPS/GF30f coincides with a slight increase of CoF from 0.38 to 0.40. PPS/CF30f shows a decreasing CoF over the duration of the test run from 0.41 to 0.26. PTFE significantly improves the friction and wear behavior of PPS matrix composites, which is shown both in a decrease in wear and CoF. In comparison to the specimen discussed above, a prolonged run-in phase is observed, which is indicated by variations of the CoF. At steady state, 0.27 and 0.17 are reached at the end of the run for PPSPTFE/GF30f and PPSPTFE/CF30f. After the run-in phase, both lubricated PPS composites show linearly accumulating wear with PPSPTFE/CF30f performing best.

Specific wear rates were calculated for the final 500 iterations, confirming a decrease in wear by one order of magnitude for the carbon fiber-reinforced PPS materials due to the presence of PTFE, as summarized in Table 7. The chosen run duration of 3000 cycles is intended as a test for material preselection, enabling the observation of run-in behavior and comparison of materials in a relative manner. Tests under real conditions representative for a compressor service lifetime will be subject to further investigations.

### 3.6. Wear Track Analysis

To analyze the interactions of the tribological partners, scanning electron microscopy (SEM) images of the wear track surface after tribological testing are presented in Figure 12. The left picture (Figure 12a), showing the sample tested against PPS/CF30f, indicates a uniform surface retaining the vertical banding of the original surface finish. The metallic counter-surface of the PPSPTFE/CF30f test run depicted in Figure 12b shows an irregular darker surface structure that is typical for tribo-layer formation, as seen in previous work [35].

Quantification of the transfer film thickness deposited on the counter surface can be achieved by the extraction of FIB lamellae and analysis under transmission electron microscope (TEM). For this purpose, cross-sections in the center of the 50 mm long and 4 mm wide wear track were extracted perpendicularly to the sliding direction. TEM images of the surface are shown in Figure 13. Both images show the grain structure of the steel on the bottom half of the image. The surface tested against PPS/CF30f is covered by a protective platinum layer (black) followed by a bright carbon layer. Both were added before the focused ion milling to prevent any damage to sensitive polymer tribo-film. In contrast, the analysis of the metallic counterpart vs. PPSPTFE/CF30f indicates an approximately 100 nm thick bright layer under the protective platinum coating, which is completely covering the steel surface.

Detailed insight into the composition of the tribo-layer could be gained by conducting EELS line-scans through the region of interest shown in Figure 12b. The results are summarized in Figure 14 with the orange line indicating the concentration of fluorine atoms and a dashed purple line for sulfur. This proves that both PPS and PTFE are deposited at the tribological interface between the composite pin and metallic surface.

## 4. Discussion and Conclusions

Fiber-reinforced PPS matrix composites were shown to be promising candidate materials for high-pressure hydrogen compressor piston ring and packing solutions due to their superior mechanical performance, providing benefits to conventional (PEEK or PTFE) polymer solutions. In this work, the influence of different types of reinforcements such as glass and carbon fibers as well as the introduction of PTFE as a dry lubricating additive was investigated regarding elasto-plastic deformation behavior by complementary thermo-mechanical analysis combined with imaging.

A heterogeneous microstructure originating from the injection-molding process could be observed in all of the investigated polymers by fiber orientation gradients (Figure 4 and Figure 5). The design of piston rings has to take such fiber orientation heterogeneity into account to avoid unwanted anisotropic deformation, leading to damage initiation and premature failure (Figure 8 and Figure 9) under cyclic loading conditions. Thermal analysis showed an operation range of PPS-based polymers limited below 90 °C (Figure 7), which is critical for gas temperature and friction heat in a reciprocating compressor. Sufficient cooling of the cylinder walls and piston rod is mandatory to ensure a sufficient lifetime of PPS polymer piston and packing rings.

Glass fibers showed an insignificant benefit for the mechanical properties of the tested polymer composites, which was confirmed by its small deviation of thermo-mechanical properties between PPS/GF30f and PPSPTFE/GF30f (Figure 6a and Figure 7a) contrarily to the TMA tests where a difference in thermal expansion could be observed (Figure 7b, Table 6). An analysis of fracture behavior (Figure 8) shows weak fiber–matrix interfaces with indications of fiber pullout, diminishing reinforcing characteristics almost independent from the addition of solid lubricant (Figure 6a and Figure 7). An advantage in wear resistance for glass fibers compared to carbon fibers can be observed in the tribological testing of PPS/GF30f and PPS/CF30f (Figure 11), which were both without PTFE lubricant. Shear deformation behavior appears independent from fiber material, equal for CF and GF-reinforced PPS, which can be observed from the LOM images of the shear fracture surfaces (Figure 10). Notably, the addition of PTFE to the matrix material shows more effect on the matrix deformation behavior in regard to matrix ductility (Figure 6b).

Carbon fibers showed in contrast to the glass fibers a significant improvement of thermo-mechanical strength (Figure 6 and Figure 7) in the PPS/CF30f configuration. The addition of PTFE leads to a degradation of mechanical properties but is necessary to meet the demands of low friction and high wear resistance (Figure 11) and subsequently reach adequate service life under operation conditions. With a PTFE additive, the formation of a polymer tribo-film could be observed on the metallic counter-surface (Figure 12 and Figure 13), which is responsible for the advanced tribological properties. The tribo-film not only reduces friction but also prevents direct contact of the PPS matrix composite with the metal, reducing the material abrasion (Figure 11) of both tribo-partners.

Tribo-film analyzed by spatially resolved elemental analysis exhibits its composition of both PTFE together with PPS (Figure 14). Calculation of the mole percent ratios using their mole weights, respectively, results in an estimated ratio of PPS to PTFE in the composite of 3:1, whereas inside the tribo-film of 1:4, a polymer deposition of PTFE together with small amounts of PPS can be observed. Furthermore, the spike in sulfur and drop in fluorine content at the free surface suggest an inhomogeneous distribution of the components throughout the film. Both phenomena are subject to further investigations.

The best candidate suitable for application as a piston and packing ring material in non-lubricated high-pressure hydrogen compressors is carbon fiber-reinforced PPS with the addition of PTFE. The PPSPTFE/CF30f material has the highest thermo-mechanical strength with excellent low friction and wear properties. PTFE is responsible for tribo-film formation and increases the long-term stability of such composites suitable for operation. PTFE also reduces mechanical properties by weakening the interfaces between carbon fibers and the PPS matrix. Therefore, the PTFE content has to be balanced between the high strength and low friction of carbon fiber-reinforced PPS polymers.

The main focus of this work was to acquire a fundamental understanding of fiber-reinforced PPS for high-pressure hydrogen reciprocating compressors. It can be concluded that carbon fiber-reinforced PPS with PTFE is a suitable base material for the given application. The implementation of this class of materials will depend on the correct choice of operating parameters, whereby PPS can be used under high mechanical loads, but only if efficient temperature control of the tribo-contact is ensured. Further investigation will focus on testing potential material candidates under real operation conditions and optimizing PTFE and CF contents of the composite for best performance.

## Figures and Tables

**Figure 1 polymers-16-00412-f001:**
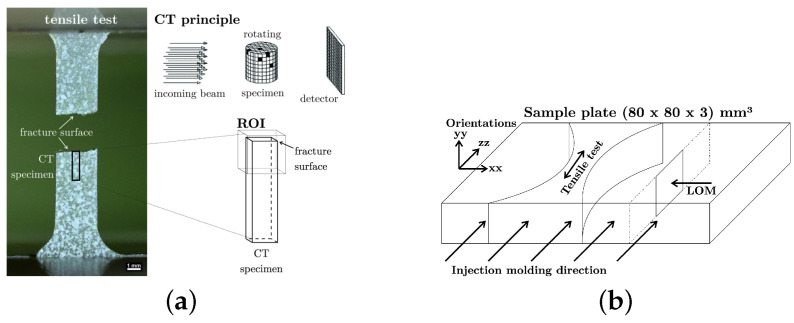
Test setup, CT region of interest and specimen location (**a**) image adapted from [26]. (**b**) Shape the of the injection-molded sample plates with the orientation of extracted tensile test specimen and ROI for microsection analyses. (**a**) Tensile test sample and region of interest for imaging. (**b**) Tensile test sample and region of interest for LOM (not to scale).

**Figure 2 polymers-16-00412-f002:**
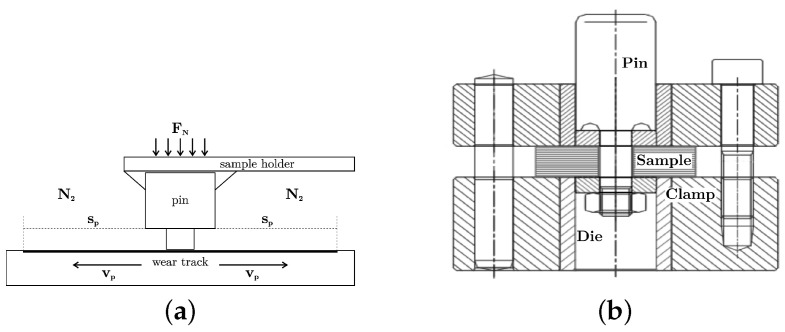
Test setup, tribology test setup schematic (**a**), image adapted from [26]. (**b**) The shear testing setup according to ISO 6721-5 [29]. (**a**) Tribology test setup. (**b**) Shear testing setup.

**Figure 3 polymers-16-00412-f003:**
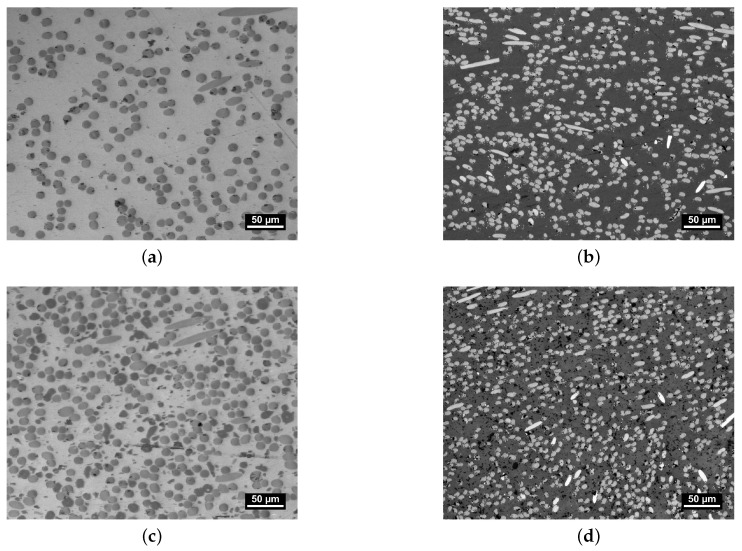
LOM, microstructure of PPS test specimen. Glass fibers (dark) are embedded in the PPS matrix (bright) in (**a**) with dark PTFE particles in (**c**). Carbon fibers (bright) embedded in the PPS matrix (dark gray) in (**b**) with dark PTFE inclusions in (**d**). (**a**) PPS/GF30f. (**b**) PPS/CF30f. (**c**) PPSPTFE/GF30f. (**d**) PPSPTFE/CF30f.

**Figure 4 polymers-16-00412-f004:**
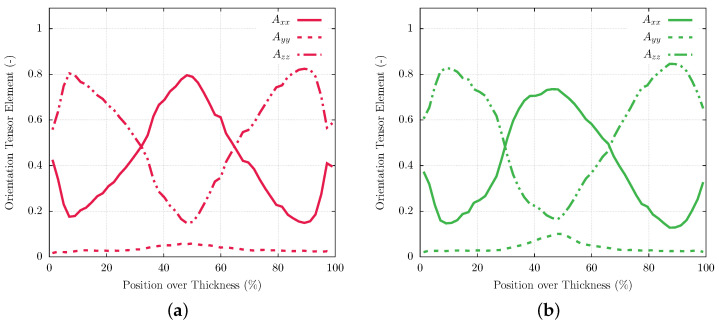
Tomography, fiber orientation tensor components in PPS/GF30f (**a**) and PPSPTFE/GF30f (**b**). Orientation of the coordinate system can be seen in Figure 1b.

**Figure 5 polymers-16-00412-f005:**
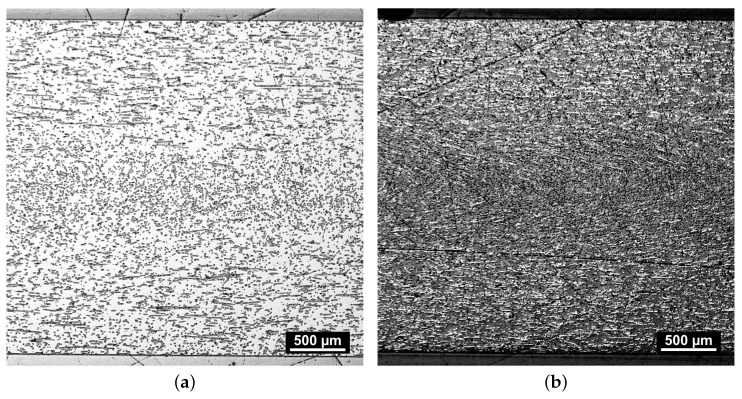
LOM, microstructure of PPS/GF30f (**a**) and PPS/CF30f (**b**) in comparison. Highly oriented fiber distribution in skin and core layers of the work piece.

**Figure 6 polymers-16-00412-f006:**
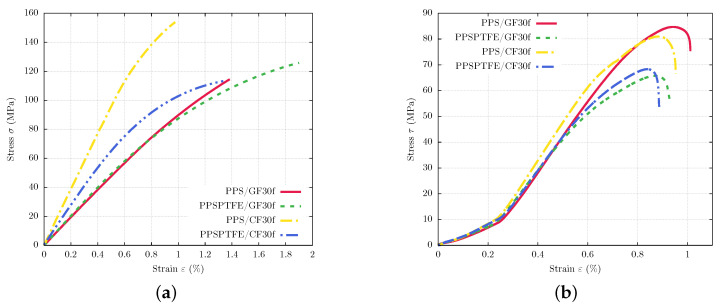
Standard testing, tensile test results at an inspection speed of 5 mm/s and sample temperature 23 °C until fracture (**left**). Shear testing results for tests according to ASTM D 732 with 1.25 mm/min crosshead speed (**right**). (**a**) Tensile test results. (**b**) Shear test results.

**Figure 7 polymers-16-00412-f007:**
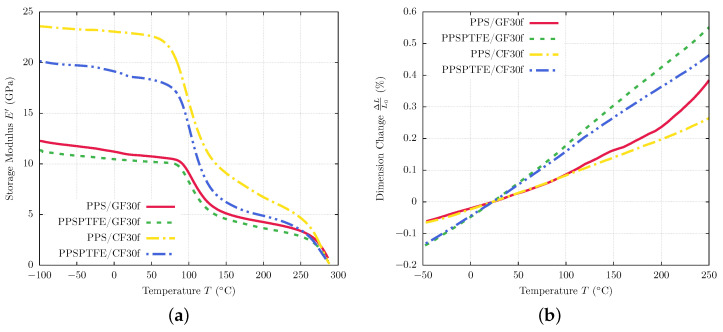
Thermal property testing, (**a**) illustrating the thermal expansion of in a temperature range of −50 to 300 °C and heating rate of 2 K/min. Dynamic mechanical analysis in a temperature range of −100 to 300 °C at a heating rate of 3 K/min in (**b**). (**a**) Dynamic mechanical analysis results. (**b**) Thermal expansion test results.

**Figure 8 polymers-16-00412-f008:**
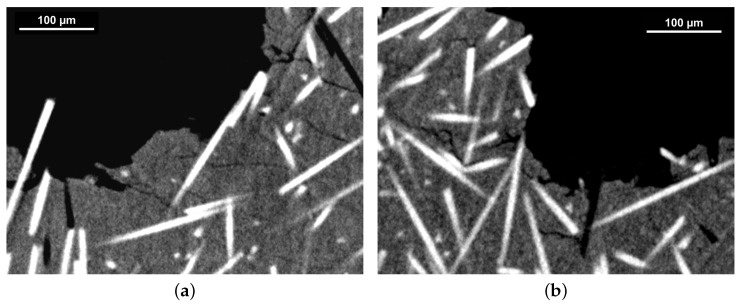
XCT, fracture surface images of PPS/GF30f (**a**) and PPSPTFE/GF30f (**b**) at a voxel edge length of ( 1 μm^3^). Bright glass fibers embedded in gray PPS polymer matrix.

**Figure 9 polymers-16-00412-f009:**
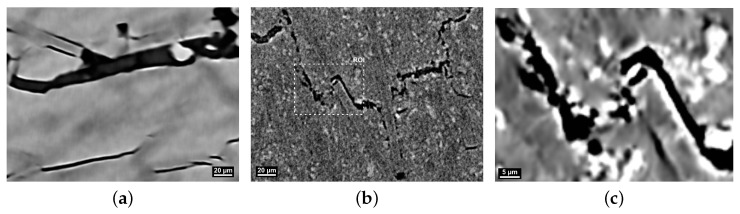
XCT, fracture surface images of PPS/CF30f (**a**) and PPSPTFE/CF30f (**b**). Darker carbon fibers embedded in gray PPS polymer matrix. ROI indicated in (**b**) shown in (**c**). Adjusted contrast settings increasing visibility of PTFE (bright) in (**b**,**c**). All images were acquired with a voxel edge length of (0.5 μm^3^). (**a**) PPS/CF30f. (**b**) PPSPTFE/CF30f. (**c**) ROI PPSPTFE/CF30f.

**Figure 10 polymers-16-00412-f010:**
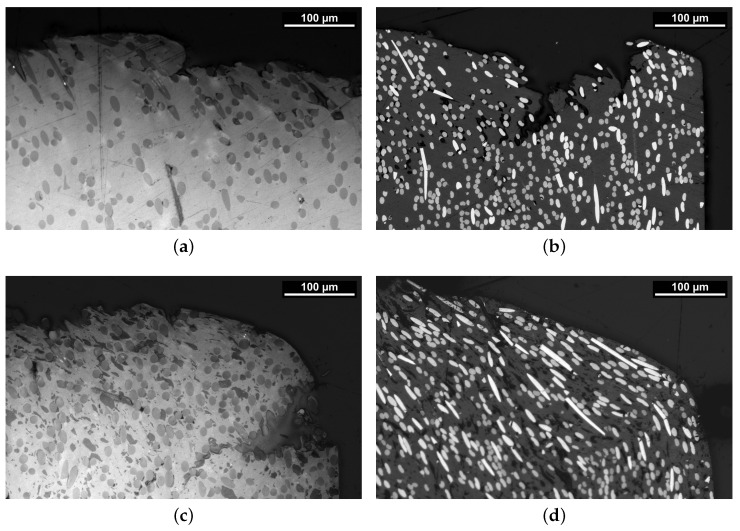
LOM, microstructure of PPS shear test fracture surfaces. Glass fibers (dark gray) are embedded in the PPS matrix (bright) in (**a**,**c**) with dark PTFE particles in (**c**). Carbon fibers (bright) embedded in the PPS matrix (dark gray) in (**b**,**d**) with dark PTFE inclusions in (**d**). (**a**) PPS/GF30f. (**b**) PPS/CF30f. (**c**) PPSPTFE/GF30f. (**d**) PPSPTFE/CF30f.

**Figure 11 polymers-16-00412-f011:**
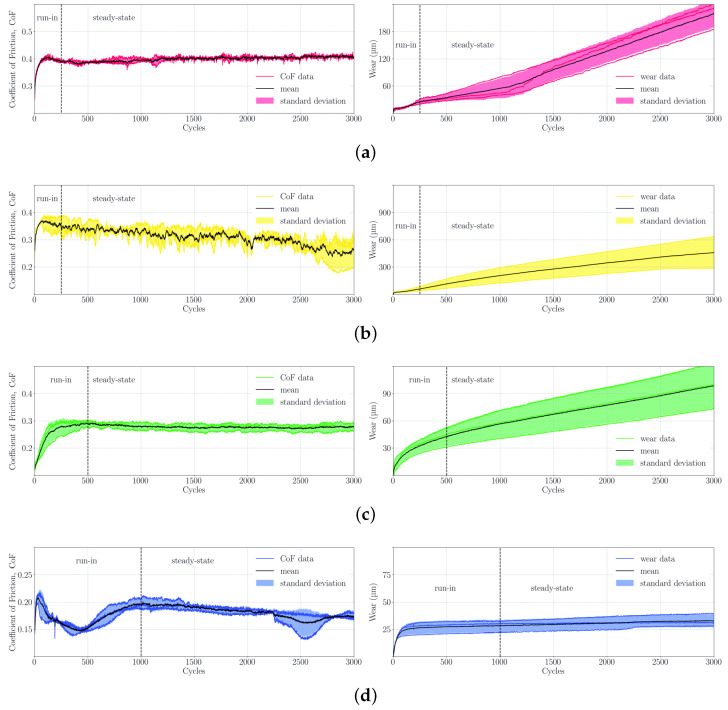
Tribology testing, coefficient of friction (CoF) and cumulative pin wear on a solution heat-treated W720 ($R_{a}=0.16\;\mu m$) counter-surface with 24.8 MPa contact pressure and 0.1 m/s sliding speed for each tested material. The black line depicts the arithmetic mean of the test data represented with colored lines. Standard deviation of CoF and cumulative wear is illustrated by the colored surfaces. (**a**) PPS/GF30f. (**b**) PPS/CF30f. (**c**) PPSPTFE/GF30f. (**d**) PPSPTFE/CF30f.

**Figure 12 polymers-16-00412-f012:**
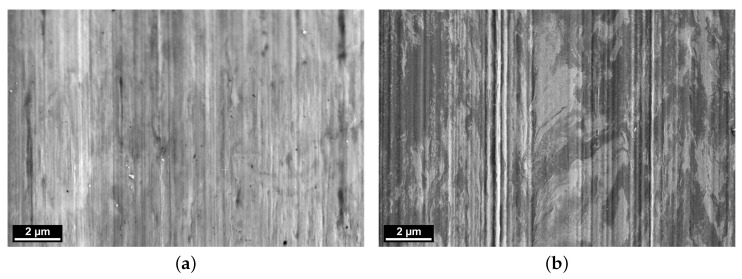
SEM, surface image of wear track on W720 tested against PPS/CF30f (**a**) and PPSPTFE/CF30f (**b**). Sliding direction is vertical with respect to image orientation. (**a**) Steel counter-surface tested against PPS/CF30f. (**b**) Steel counter-surface tested against PPSPTFE/CF30f.

**Figure 13 polymers-16-00412-f013:**
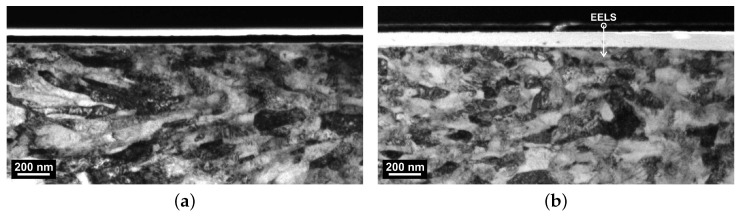
FIB, depth profile of W720 wear track tested with the PPS composites in transmission. (**a**) The W720 grain structure, covered preparation wise by protective Pt (black) and graphite (white) layers. (**b**) Polymer depositions (bright) on the metal covered by a Pt layer. Region of interest for line scan data presented in Figure 14 is shown in (**b**). (**a**) Steel counter-surface tested against PPS/CF30f. (**b**) Steel counter-surface tested against PPSPTFE/CF30f.

**Figure 14 polymers-16-00412-f014:**
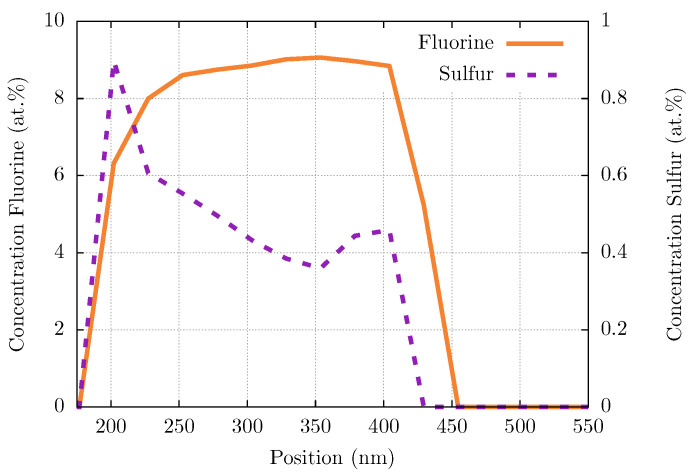
EELS, concentration of fluorine (orange, solid line) and sulfur (purple, dashed line) atoms in polymer layer from tribo-testing PPSPTFE/CF30f against W720 steel along line scan region indicated in Figure 13b.

**Table 1 polymers-16-00412-t001:** Fiber and PTFE contents of PPS composites.

Abbreviated Form	GF (wt.%)	CF (wt.%)	PTFE (wt.%)
PPS/GF30f	30	0	0
PPS/CF30f	0	30	0
PPSPTFE/GF30f	30	0	15
PPSPTFE/CF30f	0	30	15

**Table 2 polymers-16-00412-t002:** Overview of performed CT measurements and applied parameters.

Device (X-ray Target)	Mode	Voxel Edge Length Text (μm^3^)	Voltage (kV)	Integration Time (ms)	Average	Aim
Nanotom 180 NF (DW)	0 (big)	2	80	600	3	Fiber characterization
Nanotom 180 NF (Mo)	2 (small)	1	55	1200	9	Fracture surface
Easytom (LaB6)	small	0.5	60	1000	3	Fracture details

**Table 3 polymers-16-00412-t003:** Material composition of solution heat-treated W720 steel counterparts.

C (wt.%)	Si (wt.%)	Mn (wt.%)	Mo (wt.%)	Ni (wt.%)	Co (wt.%)	Ti (wt.%)	Al (wt.%)
≤0.03	≤0.1	≤0.1	5.0	18.5	9.0	0.7	0.1

**Table 4 polymers-16-00412-t004:** Mechanical tests, ultimate tensile strength and shear strength results.

Abbreviated Form	Ultimate Tensile Strength $R_{m}$ (MPa)	Shear Strength $\tau_{s}$ (MPa)
PPS/GF30f	113.2±1.8	84.7±0.7
PPS/CF30f	153.3±1.0	85.3±6.1
PPSPTFE/GF30f	123.5±4.2	65.7±0.7
PPSPTFE/CF30f	112.2±1.7	67.2±1.6

**Table 5 polymers-16-00412-t005:** Vickers hardness testing results with 5 kg test load and 30 s holding time.

Abbreviated Form	Vickers Hardness (HV)
PPS/GF30f	28.8±0.2
PPS/CF30f	33.1±0.3
PPSPTFE/GF30f	24.8±0.2
PPSPTFE/CF30f	27.2±0.5

**Table 6 polymers-16-00412-t006:** Calculated linear coefficient of thermal expansion from thermal expansion testing for a temperature range of 35 to 90 °C.

Abbreviated Form	CTE $\left(\mathrm{ppm}/K\right)$
PPS/GF30f	11.0±4.9
PPS/CF30f	10.9±2.8
PPSPTFE/GF30f	22.9±3.6
PPSPTFE/CF30f	20.8±1.6

**Table 7 polymers-16-00412-t007:** Summary of tribology testing data presenting coefficient of friction and specific wear rate were averaged over the last 500 measurements and the n=3 tests per PPS matrix material examined.

Abbreviated Form	CoF	Specific Wear Rate (mm^3^/(N m))
PPS/GF30f	0.41±0.01	$2.96\times 10^{-5}\pm 3.40\times 10^{-6}$
PPS/CF30f	0.26±0.03	$6.16\times 10^{-5}\pm 1.95\times 10^{-5}$
PPSPTFE/GF30f	0.27±0.01	$1.32\times 10^{-5}\pm 2.67\times 10^{-6}$
PPSPTFE/CF30f	0.17±0.01	$4.36\times 10^{-6}\pm 5.98\times 10^{-7}$

## Data Availability

The data presented in this study are available on request from the corresponding author. The data are not publicly available due to confidentiality.

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
