# Peer review of "Characterization of PPS Piston and Packing Ring Materials for High-Pressure Hydrogen Applications"

_polymers, 2024, doi:10.3390/polym16030412_

Round 1
Reviewer 1 Report
Comments and Suggestions for Authors
The manuscript titled "Characterization of PPS Piston and Packing Ring Materials for High-Pressure Hydrogen Applications," presented by Pöllinger et al., compares carbon and glass fiber-reinforced PPS matrix polymers in terms of the relationships between microstructures and composite properties, such as thermal properties and fracture mechanisms. While the manuscript is grounded in fruitful experimental data, some areas could be further improved, as suggested below:
1. Figures 2a and c show different density distributions of fibers; however, Table 1 reveals that PPS/GF30f and PPSPTFE/GF30f had the same loading amount of fibers. Please explain the reason for this discrepancy.
2. The authors claim that the pull-out mechanism is the main failure mechanism under tensile strength, but the interface should be important during the pull-out process. Are there any interactions or bonding between the matrix and fillers? A detailed exploration of this aspect would enhance the understanding of the observed mechanical behavior.
3. Although the database is fruitful, it is better for the authors to emphasize the main data that contributes to the conclusion. With this concern, the authors are suggested to rearrange the manuscript and make the main idea clearer to the readers.
Comments on the Quality of English LanguageNA
Reviewer 2 Report
Comments and Suggestions for Authors
The manuscript “Characterization of PPS Piston and Packing Ring Materials for High-Pressure Hydrogen Applications“ contains many serious inaccuracies that need to be clarified.
General remarks
1 It should be clarified in the article whether the rheological behavior under long-term loads was investigated.
2 Lack of basic information about glass fibers and carbon fibers such as diameter, tex, tensile strength, modulus of elasticity, ultimate strain, etc.
3 You should write in the article what are the properties of the PPS and PTFE materials used in the matrix.
4. Element stiffness is not the same as storage modulus (156 line). It should be clarified what other factor influences stiffness. Moreover, it should be explained in the article (here) whether the value of the elastic modulus was constant or variable in such a wide temperature range.
5. It should be explained in the article whether the authors took into account the specificity of CTE for glass fibers and carbon fibers and what it involves.
6. Tensile test and shear test (223 line): The authors should explain what causes the greater deformation of the glass fiber material. No greater reflection on the properties of the materials used.
7. The authors should explain in the article why they did not conduct tensile tests in other temperature ranges, which is a necessary condition for this type of use of the tested materials.
8. The authors wrote that the addition of PTFE does not have a significant effect on the tensile strength (Fig. 5). Meanwhile, the graph shows that the addition of PTFE reduces the tensile strength of the carbon fiber material. This should be explained in the article.
9. The article should include how many samples were tested in the tensile and shear test, and what the SD and COV were.
​Specific remarks
10. 73 line: the abbreviation GF should be explained.
11. 153 line: Please write the standard according to which the tensile strength was tested.
12. 166 line: a diagram or photograph of the study must be provided.
13. 232 line: What does it mean: compression tests?
14. 341 line, Fig. 12b why there is a signature b) PPSPTFE/CF30f. This needs to be clarified.
I recommend an in-depth review of the manuscript, including comments, to make it an article suitable for publication in the Polymers.
In its current state, the article should not be published.
Comments on the Quality of English LanguageMinor editing of English language required.
Round 2
Reviewer 2 Report
Comments and Suggestions for Authors
Compared to the previous version of the article, they have introduced corrections that largely take into account the reviewer's suggestions.
The article may be published.